# Chiral adiabatic transmission protected by Fermi surface topology

Isidora Araya Day[1, 2, *], Kostas Vilkelis [1, 2], Antonio L. R. Manesco [2], A. Mert Bozkurt [1, 2, †],
Valla Fatemi [3], Anton R. Akhmerov [2, ‡]

**1** QuTech, Delft University of Technology, Delft 2600 GA, The Netherlands
**2** Kavli Institute of Nanoscience, Delft University of Technology, P.O. Box 4056, 2600 GA Delft,
The Netherlands
**3** School of Applied and Engineering Physics, Cornell University, Ithaca, NY 14853 USA
* iarayaday@gmail.com † a.mertbozkurt@gmail.com ‡ cat@antonakhmerov.org

November 28, 2023

## Abstract

We demonstrate that Andreev modes that propagate along a transparent Josephson junction have a perfect transmission at the point where three junctions meet. The chirality and the number of quantized transmission channels is determined by the topology of the Fermi surface and the vorticity of the superconducting phase differences at the trijunction. We explain this chiral adiabatic transmission (CAT) as a consequence of the adiabatic evolution of the scattering modes both in momentum and real space. We identify an effective energy barrier that guarantees quantized transmission. We expect that CAT is observable in non-local conductance and thermal transport measurements. Furthermore, because it does not rely on particle-hole symmetry, CAT is also possible to observe directly in metamaterials.

Unlike particles that follow deterministic trajectories, waves, both quantum and classical, may split and follow multiple paths. Under special conditions, however, waves follow a deterministic path, transmitting perfectly from source to target. The simplest mechanism that protects such transmission is the adiabaticity of the potential landscape: if the potential changes slowly enough, the wave functions adjust to the local changes of the potential without splitting into partial waves. In a quantum point contact [1], for example, the adiabaticity of the constriction ensures that an integer number of modes pass through, while the rest of the modes reflect. Another mechanism that protects quantized transmission is the topology of a gapped Hamiltonian, which prohibits scattering between channels due to a combination of their symmetry structure and spatial separation. For example, the chiral edge transport of a quantum Hall insulator [2] is protected because the channels propagating in opposite directions occupy different edges of the sample, and are separated by a gapped bulk.

Topological protection, however, extends beyond the bulk properties of an insulator. Specifically, the number of electron- and hole-like Fermi surfaces give rise to the quantized transmission of Andreev modes propagating in a superconductor – normal metal – superconductor (SNS) junction at a $\pi$ phase difference [3]. While these modes are dispersionless within the Andreev approximation (the linearization of the Hamiltonian at the Fermi level) they acquire charge and velocity due to the nonlinearity of the normal dispersion. At positive voltage bias, the nonlocal conductance measures the number of electron-like *critical points*: Fermi surface points where the velocity is parallel to the interface between the superconductors. Likewise, negative bias conduc-

tance counts the number of hole-like critical points. The difference between electron and hole-like critical points is the Euler characteristic of the Fermi surface—a topological invariant [4].

To highlight our main result, we refer the reader to Fig. 1: a multiterminal short SNS junction has quantized chiral transmission of Andreev modes (see App. A and Ref. [5] for the details of numerical simulations). The SNS junctions have a phase difference unequal to $\pi$, so that the chiral transmission only exists above a minimal energy, and therefore the transmission is not protected by a symmetry. While protected chiral transport also exists in quantum Hall systems, the modes in the SNS junction occupy the same spatial region, and therefore the mechanism is distinct. In the following, we examine this phenomenon and explain how chiral transmission emerges from the dispersion of the Andreev states beyond the Andreev approximation and the topology of the Fermi surface. Because the scattering is protected by the adiabaticity of the wave function evolution, we name this phenomenon *chiral adiabatic transmission* (CAT).

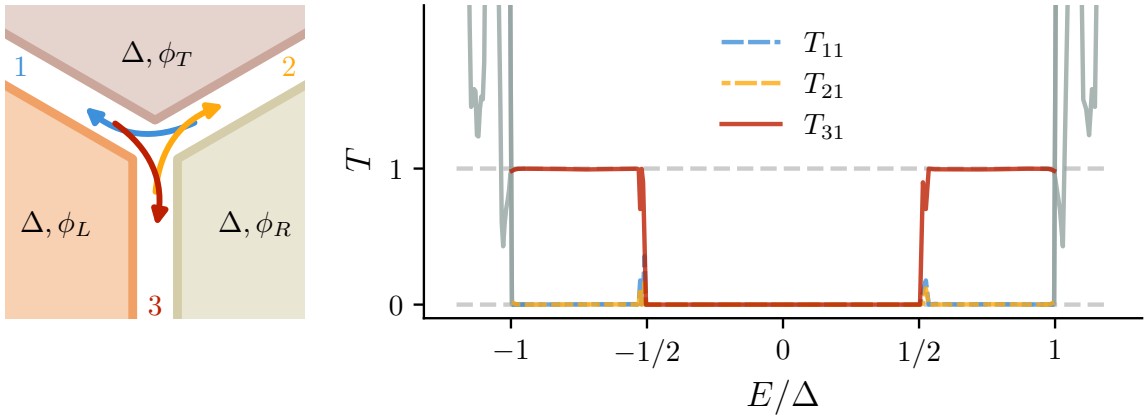

Figure 1: A three-terminal Josephson junction has quantized chiral transmission of Andreev modes. Left: Three superconductors with an infinitesimally narrow ballistic metal in between. The superconductors have the same normal Hamiltonian with chemical potential $\mu$. The gap $\Delta$ is constant, but the superconducting phases $\phi_T$, $\phi_L$, and $\phi_R$ differ, such that all the phase differences are $2\pi/3$. Andreev modes propagate along the junctions as shown by the arrows. Right: Transmission of Andreev modes from lead 1 into itself ($T_{11}$), lead 2 ($T_{21}$), and lead 3 ($T_{31}$), for a three-fold symmetric trijunction.

To understand the origin of CAT, we consider a Josephson junction: a normal metal between two superconductors, as shown in Fig. 2(a). We generalize the result of Ref. [3] to an arbitrary phase difference between the superconductors, as necessary to analyze a trijunction. To identify the role of corrections to the Andreev approximation, we consider a parabolic dispersion in the direction $y$, perpendicular to the interface between the superconductors. This is a good approximation close to a critical point and still gives a qualitatively valid description of the dispersion away from the critical point. We consider two s-wave superconductors with a gap $\Delta$ and a phase difference $\delta\phi$, with an infinitesimally narrow metal between them. At a fixed momentum $k_x$ parallel to the junction, the Bogoliubov-de Gennes Hamiltonian reads:

$$H(y) = [-a\partial_y^2 - E_x]\tau_z + \Delta\cos(\delta\phi/2)\tau_x + \text{sign}(y)\Delta\sin(\delta\phi/2)\tau_y, \tag{1}$$

where $\tau_{x,y,z}$ are Pauli matrices acting on the particle-hole degree of freedom, $2a$ is the inverse effective mass, and $E_x$ is the position of the band bottom. Considering the dispersion near a

critical point with $k_x = k_c + \delta k_x$ gives $E_x = -v_x \delta k_x$, and reproduces the Hamiltonian of Ref. [3] when $\delta \phi = \pi$. For a parabolic band, on the other hand, $E_x = \mu - a k_x^2$, with $\mu$ the chemical potential.

The Andreev approximation follows from linearizing the dispersion around the Fermi momentum $k_{y0} = \pm\sqrt{E_x/|a|}$ in the $y$-direction and using the Ansatz $|\Psi(y)\rangle = \exp(i k_{y0} y)|\psi\rangle$, where $|\psi\rangle$ is a two-component spinor that only changes slowly with $y$. This approximation is valid when $\Delta \ll E_x$. Applying Eq. (1) to $|\Psi(y)\rangle$ and neglecting $\partial_y^2|\psi\rangle$, we find that $|\psi\rangle$ is an eigenstate of the linearized Hamiltonian

$$H_\pm^{(0)}(y) = \mp 2i a k_{y0} \partial_y \tau_z + \Delta \cos(\delta\phi/2)\tau_x + \mathrm{sign}(y)\Delta \sin(\delta\phi/2)\tau_y. \tag{2}$$

This Hamiltonian has one bound state for each sign of $\pm k_{y0}$, which we use to construct the approximate eigenstates of $H(y)$:

$$|\Psi_\pm^{(0)}(y)\rangle = \sqrt{\frac{\Delta \sin|\delta\phi/2|}{v_y}}\begin{pmatrix} \pm 1 \\ 1 \end{pmatrix}\exp\left[\pm i k_{y0} y - \frac{\Delta \sin|\delta\phi/2|}{v_y}|y|\right], \tag{3}$$

where $v_y = 2a k_{y0}$. The corresponding eigenvalues $E_\pm^{(0)} = \pm\Delta\cos(\delta\phi/2)$ are the result of the Andreev approximation. To go beyond the linear approximation, we project the full Hamiltonian $H(y)$ onto the basis states $|\Psi_\pm^{(0)}(y)\rangle$, keep only terms up to $\mathcal{O}(\Delta^2)$, and obtain the effective Hamiltonian

$$H_\pm = \Delta\begin{pmatrix} \cos(\delta\phi/2) & \frac{\Delta}{2E_x}\sin^2(\delta\phi/2) \\ \frac{\Delta}{2E_x}\sin^2(\delta\phi/2) & -\cos(\delta\phi/2) \end{pmatrix}. \tag{4}$$

This yields the dispersion of the Andreev modes

$$E_\pm = \pm\Delta\sqrt{(\Delta/2E_x)^2\sin^4(\delta\phi/2) + \cos^2(\delta\phi/2)}, \tag{5}$$

and the corresponding eigenstates

$$N_\pm|\Psi_\pm(y)\rangle = \left(\cos(\delta\phi/2) \pm \frac{E_\pm}{\Delta}\right)|\Psi_+^{(0)}(y)\rangle + \frac{\Delta}{2E_x}\sin^2(\delta\phi/2)|\Psi_-^{(0)}(y)\rangle, \tag{6}$$

where $N_\pm$ is a normalization factor. The relative weights of the momenta $\pm k_{y0}$ contributed by $|\Psi_\pm^{(0)}(y)\rangle$ depend on $\delta\phi$ and $E_x/\Delta$, and are only equal at $\delta\phi = \pi$. Away from $\delta\phi = \pi$, the Andreev modes are asymmetric superpositions of the states at $\pm k_{y0}$, and the average momentum of the Andreev modes is misaligned with the junction. Figure 2(b) shows the orientation $\theta = \arctan(k_x/\langle k_y\rangle)$ of the average Andreev mode momentum as a function of $k_x$. At the lowest available energy, the Andreev modes are perpendicular to the interface, while at the critical points, the Andreev modes align with the junction's direction.

In a trijunction, the momentum changes adiabatically because superconducting pairing—the only position-dependent Hamiltonian term—is small. Therefore, to analyze the scattering, we consider the energy of the different states as a function of their momentum expectation value. Within each junction, the state with the highest energy is the one with momentum parallel to the junction, see Eq. (5). This provides an effective energy barrier that separates scattering states in momentum space. Due to this barrier, the adiabatic evolution of momentum prohibits the instantaneous momentum value from being parallel to the junction. In Fig 3, we sketch the energy of the Andreev modes in each of the three arms as a function of the orientation of average state

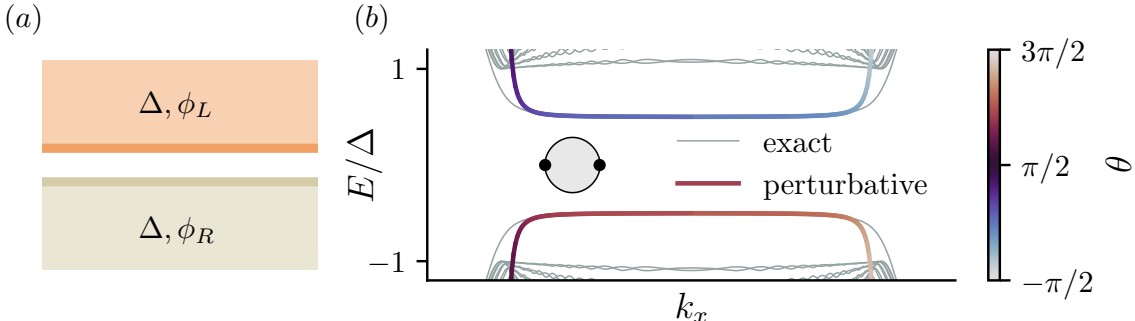

Figure 2: The quadratic dispersion of the Andreev modes hybridize states at opposite momenta, enabling the propagation of the Andreev modes along the junction. (a) SNS short junction with arbitrary $\phi_L - \phi_R$ phase difference. (b) Spectrum of the junction in (a) for a normal dispersion with circular Fermi surface (inset), two critical points (black dots), and $\phi_L - \phi_R = 2\pi/3$. The perturbative dispersion of the Andreev modes (5) is colored according to the angle of the momentum expectation value $\theta$.

momentum. The energy barrier prohibits some scattering processes while allowing others. For example, a state incoming from lead 3 has three possible scattering trajectories: it can either propagate to lead 1, 2, or reflect into lead 3. A trajectory that propagates into lead 2 without encountering lead 1 does not cross any energy barriers because its momentum does not align with any of the junctions. On the other hand, for the same state to scatter into lead 1, it must cross both an energy barrier due to the lead 3, and lead 1. Altogether, the energy barriers provided by different leads only allow the scattering processes that have the same chirality. This separation of modes in momentum space is reminiscent of the mechanism protecting quasi-Majorana modes: approximate zero modes appearing in topologically trivial superconductors with broken time-reversal symmetry in presence of smooth confining potentials [6–8].

Our arguments rely exclusively on the angle dependence of the energy barrier, and the adiabatic evolution of the momentum. Therefore, it is natural to expect that CAT does not depend on the details of the junction, the shape of the Fermi surface, or even the presence of particle-hole symmetry. To confirm this assumption, we simulate a trijunction with the following modifications:

- The phase differences across the three junctions are unequal.

- The junction has unequal angles between its arms.

- The Fermi surface is anisotropic.

- Particle-hole symmetry is artificially broken.

The resulting transmissions are shown in Fig. 4. The only qualitative difference from the symmetric trijunction is that the different channels are open at different energy ranges due to the different phase differences. We observe that despite these modifications, the transmission stays quantized and chiral.

To prove that CAT is protected by the topology of the Fermi surface, rather than the number of open channels, we consider a model with a next-nearest neighbor hopping in the $y$-direction, such that it has a peanut-shaped Fermi surface. The resulting transport simulations are shown in

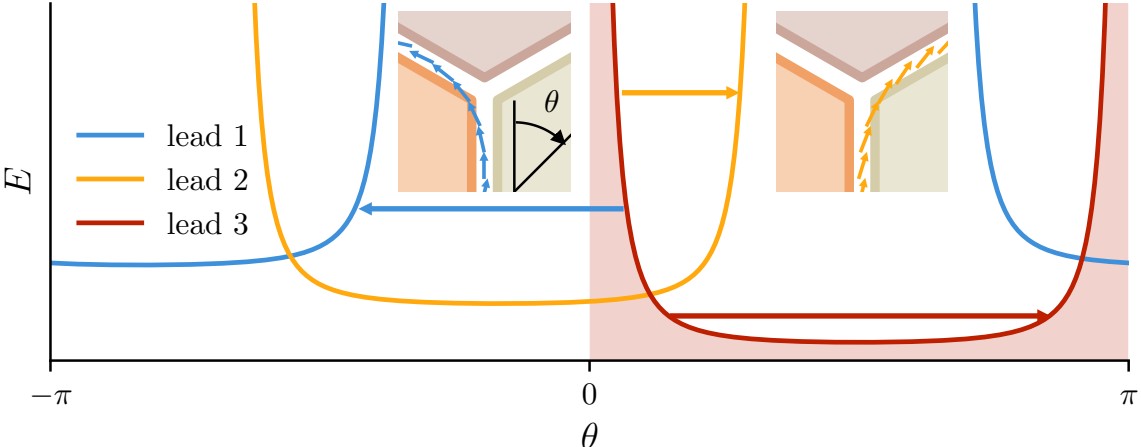

Figure 3: The energy of a mode in each of the three arms of the trijunction, as a function of its orientation $\theta$. Each arm has a different $\delta\phi$, so that the minimal energy depends on the arm. This energy defines an effective barrier (shaded area) that prevents scattering processes whose momentum aligns with any of the junctions (blue arrow), while permitting scattering processes whose momentum does not (yellow and red arrows). The local expectation value of momentum along the scattering path aligns with at least one arm of the junction for a prohibited scattering process (left inset) and does not for an allowed scattering process (right inset).

Fig. 5. The additional critical points of the Fermi surface that appear in two out of three arms of the trijunction create extra particle-like and hole-like channels [3]. At the trijunction the additional channels couple in a way that is sensitive to the junction shape and may either reflect or partially transmit. Despite that, examining the individual transmission eigenvalues—eigenvalues of $t_{ji}t_{ji}^{\dagger}$ with $t_{ji}$ the transmission matrix from lead $i$ to $j$—reveals that one of the eigenvalues stays quantized and chiral. This follows from the structure of the energy barriers created by the mode hybridization: at least one of the channels is always prevented from any scattering process other than chiral transmission.

So far we focused on the transmission of Andreev modes, without considering the electrical conductance. Our work differs from Ref. [3] in that we consider finite phase differences, and therefore do not rely on time-reversal symmetry. On the other hand, the electrical conductance in Ref. [3] is quantized because it is impossible to couple opposite sides of the Fermi surface in the absence of large momentum scattering. To confirm the robustness of the electrical conductance quantization, we simulate an interface between a Josephson junction and a normal lead, and compute the transmissions from the Andreev mode to the electron modes $T_{ea}$ and hole modes $T_{ha}$, with the result being shown in Fig. 6(a). The electrical conductance in a symmetric NSN geometry equals to $(e^2/h)T_{ea}(T_{ea} - T_{ha})$. We observe that similarly to Ref. [3], the Andreev mode perfectly couples to electron modes at positive bias voltage, and hole modes at negative bias voltage.

The coupling between the two approximate eigenstates $|\Psi_{\pm}^{(0)}\rangle$ is $\propto \Delta^2/E_x$, and it is similar to the energy $\Delta^2/\mu$ of a Caroli-de Gennes-Matricon (CdGM) bound state in a superconducting vortex [9], where $\mu$ is the distance between the band bottom and the Fermi level. This similarity is not accidental: like the Andreev modes, the momentum distribution of the CdGM states is confined

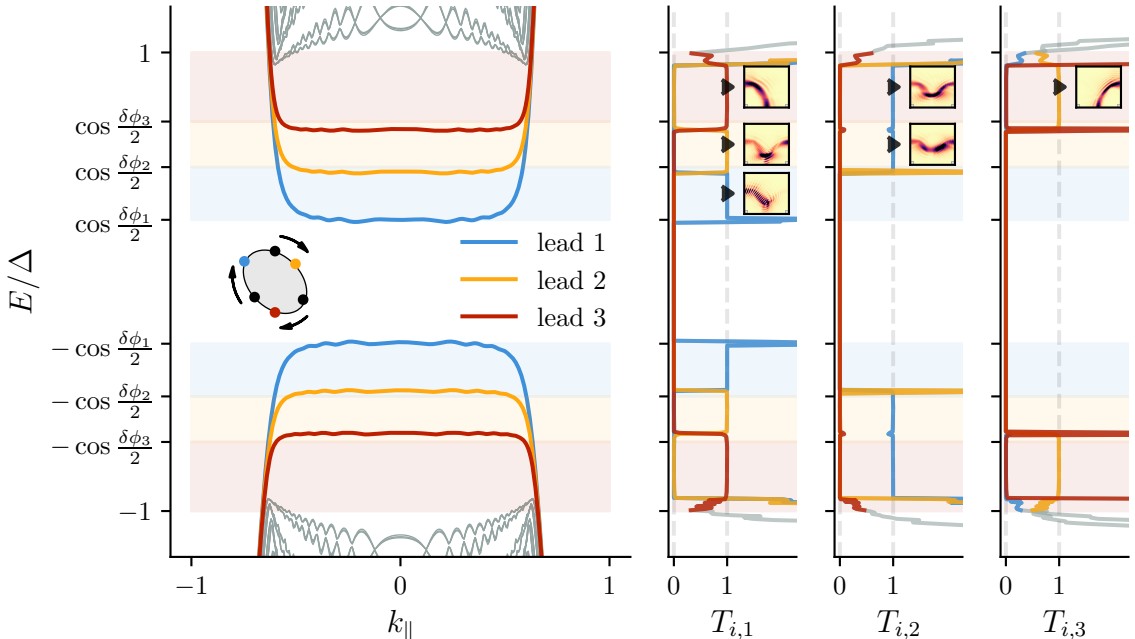

Figure 4: Dispersion and quantized chiral transmission of Andreev modes in an asymmetric trijunction. Left: Dispersion of each lead along the junction's direction. The leads have the same normal Hamiltonian with an anisotropic Fermi surface (inset), but the critical points are at different momenta, schematically shown with the dots. Right: Transmission of Andreev modes from leads 1, 2, and 3, respectively, into the other leads or themselves. The density of the scattering wave function originating from each lead is shown in the insets for different energies (black triangles). The colors label the outgoing leads.

to a cross-section of the Fermi surface and their wave functions possess a similar electron-hole asymmetry. In Fig. 6(b), we show that transmission from a CdGM mode to electrons and holes has the same quantized conductance as that of the Andreev modes. Differently from Andreev modes, however, higher energy CdGM modes contribute the same conductance as the lowest energy mode, so that the total number of quantized conductance channels in a vortex is proportional to $\mu/\Delta$. The conductance quantization of CdGM modes, together with the unexplained quantization of the rectified conductance in a superconducting quantum point contact [4] hints at a more universal description of the underlying protection.

An experimental observation of CAT requires a ballistic Josephson junction. While we considered a position-independent normal Hamiltonian, we expect that a sufficiently smooth potential landscape will not affect the transmission. Thus, candidate platforms must have high mobility and smooth normal-superconductor interfaces, potentially realizable in several platforms. Devices with these properties have been fabricated using two-dimensional electron gases [10–12] and stackings of graphene with superconducting transition metal dicalchogenides [13–15]. Alternatively, twisted bilayer graphene and Bernal bilayer graphene offer gate-defined Josephson junctions with intrinsic superconductivity tunable by electric fields [16–19]. The ability to measure nonlocal electrical conductance while the superconductors are grounded poses an additional

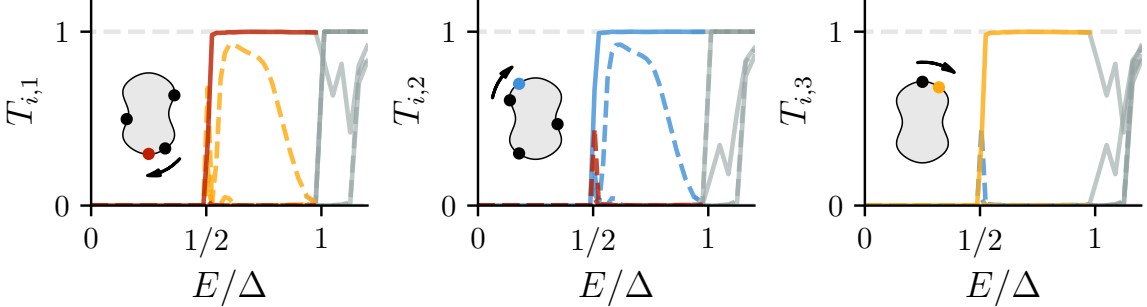

Figure 5: CAT in a trijunction with a peanut-shaped Fermi surface, with Euler characteristic $\chi_F = 1$ equal to that of a circular Fermi surface. The panels show the transmission eigenvalues of Andreev modes from lead 1, 2, and 3, respectively, into the other leads, while reflections are not shown. Only one of the eigenvalues per panel is quantized and chiral (solid line), process shown by the arrows from critical points in the incoming lead (black dots) to the critical points in the outgoing lead (colored dots). Unprotected transmissions are shown in dashed lines.

challenge to observe the imprint of the Fermi surface topology on the Andreev transport. To suppress the contribution of the supercurrent to nonlocal conductance, many experiments operate in the tunneling regime [10,20–22], which breaks the quantization of nonlocal conductance. On the other hand, because CAT produces asymmetry of transmission rather than only quantization, it becomes easier to observe. For instance, in addition to purely 2D systems, we expect chiral transport to manifest in high quality films of crystalline superconductors such as aluminum, where the Josephson junctions are formed by narrowing the film thickness. Finally, the chiral nature of the transport makes it observable in thermal transport measurements, which are less sensitive to supercurrent.

Metamaterials offer another platform to observe CAT. Because introducing phase differences requires breaking time-reversal symmetry, single valley transport in photonic or acoustic honeycomb crystals [23–26] is a promising starting point. In such a system coupling an electron-like and a hole-like band that coexist in a single valley mimics the effect of the superconducting pairing. A displacement of the valleys in momentum space then shifts the relative phase difference, implementing an analog of the superconducting phase difference. In addition to microscopic control over the effective Hamiltonian, metamaterials naturally allow local high resolution probes and therefore make the chiral nature of the scattering modes directly observable.

In summary, we have analyzed the transport of Andreev modes in a three terminal Josephson junction. We demonstrated that the Fermi surface topology and adiabaticity enable quantized chiral transmission by separating different channels in momentum space. The chiral nature of the transport makes it observable in thermal, rather than only electrical, transport measurements. Furthermore, because the transmission only relies on the adiabaticity, rather than particle-hole symmetry, this phenomenon is also observable in metamaterials. That the same phenomenology applies to superconducting vortices suggests a more general underlying description, which we leave for future work.

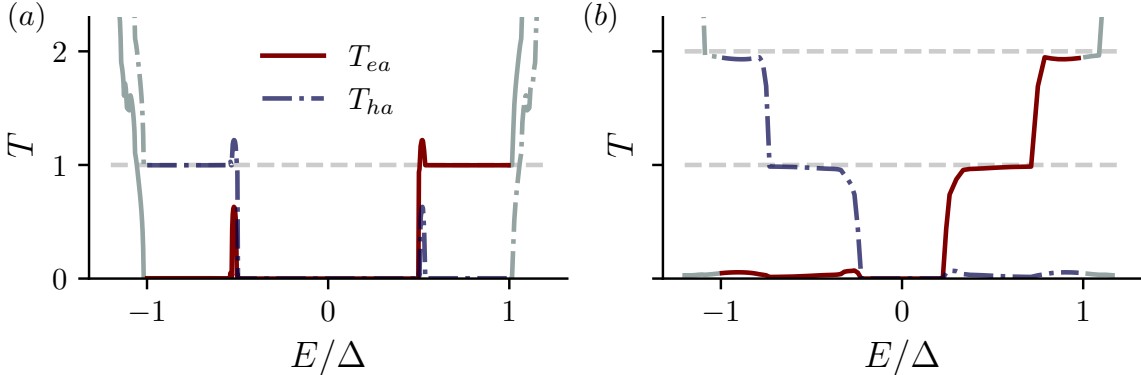

Figure 6: Quantization of coupling between Andreev modes and electrons and holes at a normal metal – Josephson junction interface. (a) Transmission of Andreev modes into electron and hole modes from a Josephson junction lead into a metallic one. (b) Transmission of Andreev modes into electron and hole modes from a superconducting vortex with a finite core into a metallic lead.

## Acknowledgements

We are grateful to T. Vakhtel for insightful discussions. We thank A. Bordin and A. Young for useful discussions regarding the experimental implementation of the trijunction.

## Data availability

The code used to produce the reported results and the generated data are available on Zenodo [5].

**Author contributions**   I. A. D., K. V., A. M., and A. A. performed the numerical simulations. I. A. D., K. V., and A. A. prepared the figures. I. A. D., K. V., A. M., and A. A. wrote the manuscript with input from M. B. and V. F. All authors analyzed the results and participated in defining the project scope. A. A. oversaw the project.

**Funding information**   This work was supported by the Netherlands Organization for Scientific Research (NWO/OCW) as part of the Frontiers of Nanoscience program, an NWO VIDI grant 016.Vidi.189.180, and OCENW.GROOT.2019.004. We also acknowledge funding from the European Research Council (ERC) under the European Union's Horizon 2020 research and innovation program grant agreement №828948 (AndQC).

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

# A Numerical simulations

The content in the figures of this paper was computed simulating the following tight-binding Hamiltonian using Kwant [27]:

$$H = H_{\text{superconductor}} + H_{\text{normal}} \tag{7}$$

$$H_{\text{superconductor}} = \sum_{\boldsymbol{n}} \left( \Delta e^{i\phi_{\boldsymbol{n}}} c^{\dagger}_{\boldsymbol{n},\uparrow} c^{\dagger}_{\boldsymbol{n},\downarrow} + \Delta e^{-i\phi_{\boldsymbol{n}}} c_{\boldsymbol{n},\downarrow} c_{\boldsymbol{n},\uparrow} \right), \tag{8}$$

$$H_{\text{normal}} = \sum_{\sigma=\uparrow,\downarrow} \sum_{\boldsymbol{n}} \left[ \left( 2t_x + 2t_y - \mu \right) c^{\dagger}_{\boldsymbol{n},\sigma} c_{\boldsymbol{n},\sigma} - \left( t_x c^{\dagger}_{\boldsymbol{n}+\boldsymbol{e}_x,\sigma} c_{\boldsymbol{n},\sigma} + t_y c^{\dagger}_{\boldsymbol{n}+\boldsymbol{e}_y,\sigma} c_{\boldsymbol{n},\sigma} + \text{h.c.} \right) \right], \tag{9}$$

where $c^{\dagger}_{\boldsymbol{n},\sigma}$ ($c_{\boldsymbol{n},\sigma}$) creates (annihilates) an electron with spin $\sigma$ at site $\boldsymbol{n} = (n_x, n_y)$ on a square lattice. The superconducting phase $\phi_{\boldsymbol{n}}$ is site-dependent, while the superconducting gap $\Delta$, chemical potential $\mu \in \mathbb{R}$, and the hopping amplitudes $t_x, t_y$ are uniform.

To compute the transmission in the three-fold symmetric trijunction of Fig. 1 we use a square lattice of size $L = 100$, with parameters $\mu = 0.5$, $\Delta = 0.1$, $t_x = t_y = 1$, and phases $\phi_L = 2\pi/3$, $\phi_R = -2\pi/3$, and $\phi_T = 0$ for the left, right, and top regions respectively. We use the same parameters to compute the band structure in Fig. 2. The system for the trijunction is shown in Fig. 7, and the code used to compute the transmission is available at [5], together with the code used to generate all the other figures in this paper.

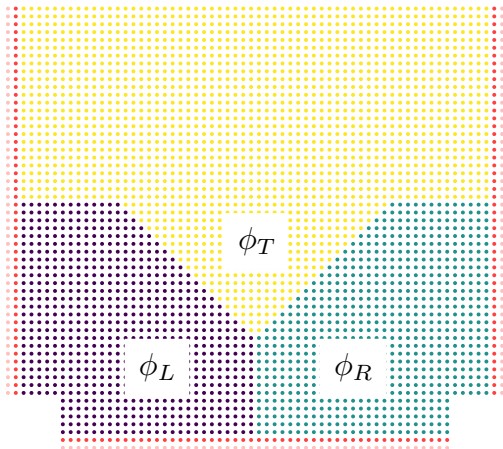

Figure 7: The trijunction geometry. The system is composed of three superconducting regions (left, right, and top) of superconducting phase $\phi_L$, $\phi_R$, and $\phi_T$, respectively. The left and right leads (red) are periodic in the x-direction, and the bottom lead (red) is periodic in the y-direction.

In Fig. 4 we make Fermi surface anisotropic by adding the diagonal hoppings to the Hamiltonian:

$$H_{\text{anisotropy}} = \sum_{\sigma=\uparrow,\downarrow} \sum_{\boldsymbol{n}} \left[ 2t_{xy} c^{\dagger}_{\boldsymbol{n},\sigma} c_{\boldsymbol{n},\sigma} - \left( t_{xy} c^{\dagger}_{\boldsymbol{n}+\boldsymbol{e}_x+\boldsymbol{e}_y,\sigma} c_{\boldsymbol{n},\sigma} + \text{h.c.} \right) \right]. \tag{10}$$

To compute the spectrum and transmissions in Fig. 4, we use a square lattice of size $L = 100$, with parameters $\mu = 0.5$, $\Delta = 0.1$, $t_x = t_y = 1$, and phases $\phi_L = 2\pi/3 + 1/2$, $\phi_R = -2\pi/3$, and $\phi_T = 0$ for the left, right, and top regions respectively. Moreover, we double the electron block of the Hamiltonian $H_{ee} \rightarrow 2H_{ee}$ to artificially break particle-hole symmetry, and we change the angle between the left and right arms of the junction to $\beta = \pi/2$.

In Fig. 5 we introduce the peanut-shaped Fermi surface by adding the next-nearest neighbor hoppings to the Hamiltonian:

$$H_{\text{peanut}} = -\sum_{\sigma=\uparrow,\downarrow} \sum_{n} \left[ t_{yy} c^{\dagger}_{n+2e_y,\sigma} c_{n,\sigma} + \text{h.c.} \right]. \tag{11}$$

To compute the transmissions in Fig. 5, we use a square lattice of size $L = 100$, with parameters $\mu = 0.5$, $\Delta = 0.1$, $t_x = 1.2$, $t_y = 0.8$, $t_{xy} = 0$, $t_{yy} = -0.5$, and phases $\phi_L = 2\pi/3$, $\phi_R = -2\pi/3$, and $\phi_T = 0$ for the left, right, and top regions respectively.

To compute the electrical conductance between a normal region and a Josephson junction in Fig. 6(a) we use a square lattice of size $L = 60$, and uniform parameters $\mu = 0.5$ and $t_x = t_y = 1$ for the whole system. In the superconducting regions we use $\Delta = 0.1$ and a phase difference $\phi_L - \phi_R = \pi/3$. To compute the electrical conductance between a normal region and a superconducting vortex in Fig. 6(b) we set up a three-dimensional system with additional nearest-neighbor $t_z$ hoppings in the z-direction and an onsite potential of $2t_z$. We use a lattice of size $L = 30$ and uniform parameters $\mu = 0.9$ and $t_x = t_y = t_z = 1$ for the whole system. In the superconducting region the superconducting phase forms a vortex with $\phi = \arctan(y/z)$, and the superconducting gap is position-dependent, $\Delta(y,z) = 0.25 \times \tanh(\sqrt{y^2 + z^2}/5)$.