# Peer review of "Chiral adiabatic transmission protected by Fermi surface topology"

_SciPost Physics_

## Round 1 · Referee Report · Anonymous (Referee 1) · 2024-2-5

Strengths

1- The paper is, technically, a non trivial generalization of Ref. [3].

Weaknesses

1- Very difficult to follow; toot little elements are given to understand the results contained in the 6 figures.

2- It is not clear to me whether a trijunction gives more information on the topology of the Fermi surface with respect to the single junction of Ref. [3].

Report

The paper addresses transport of Andreev states in a Josephson trijunction finding quantized transmission between normal leads. The authors relate this result to the topology of the Fermi surface. The paper is a generalization of Ref. [3], where a single Josephson junction was considered.
I find the paper not clearly written and very difficult to follow, although the results seem interesting. The paper does not meet the first three general acceptance criteria of SciPost Physics.

Requested changes

1- The concept of energy barrier is unclear. It is related to Eq. (5), but more details are needed. Fig. 3 is also unclear: are the energies of Eq. (5) plotted? What represent the barriers? Why only the area below the red curve?

2- In Fig. 4: why the second panel is different from the right panel in Fig. 1? Why also T_{1,1} and T_{2,1} are different from 0? I was expecting only T_{3,1} to be quantized.

3- In Fig. 5, what is the meaning of unprotected transmissions (dashed lines)? What are the gray dashed lines?

4- Fig. 6 refers to a single Josephson junction? What are exactly T_{ea} and T_{ha}?

5- Many captions do not contain enough information to read the figures. For example: grey lines in Fig. 1 (right) and Fig. 5; values of k_x for the critical points in Fig. 2(b); ...

---

## Round 1 · Referee Report · Anonymous (Referee 2) · 2024-5-5

Strengths

  1. Prediction of an interesting phenomenon: the perfect chiral transmission of Andreev modes in a three-terminal Josephson junction due to the topology of the Fermi surface and adiabatic transmission.
  2. Rigorous treatment together with an intuitive explanation of the results.
  3. Generally well written.

Weaknesses

  1. The only weakness is that the readability of the figures and their captions could be improved.

Report

The manuscript extends the work of Tam and Mele of Ref. 3 to the case of arbitrary superconducting phase differences. Furthermore, by considering a three-terminal Josephson junction geometry, it predicts perfect chiral transmission between the normal leads at finite energy. The phenomenon of Chiral Adiabatic Transmission (CAT) predicted in the manuscript is certainly of great interest to both the mesoscopic-transport and the superconductivity communities. The authors also discuss possible platforms for the experimental observation of CAT.

I am extremely confident that the results are valid.

The manuscript is in general well written. In particular, I found the introduction very informative and useful.

In conclusion, the manuscript easily meets the criteria for publication in SciPost Physics and I recommend publication.

My only minor suggestions regard the figures which could be made easier to read, as detailed in the list of suggested changes.

Requested changes

This is a list of suggested (not requested) changes which would make the figures (and hence the paper) more easily understandable.

  1. In Fig.2 (a), show the coordinate system x - y and a picture elucidating the definition of the angle \theta.

  2. In Fig. 3 in the inset, add the labels for the normal leads (1,2,3).

  3. In the insets of the left panel of Fig 4. the probability densities are not readable. I would suggest either to remove them or to show them in a separate figure together with an outline of the geometry.

  4. It took me some time to realise that in Figs 1-2 and 4-6, the results for states with energies outside the gap are depicted in a different colour (cyan). It would be really helpful to state this explicitly.

Recommendation

Publish (easily meets expectations and criteria for this Journal; among top 50%)

---

## Editorial Decision

resubmitted